# “*What Bothers Me Most Is the Disparity between the Choices that People Have or Don’t Have*”: A Qualitative Study on the Health Systems Responsiveness to Implementing the Assisted Decision-Making (Capacity) Act in Ireland

**DOI:** 10.3390/ijerph17093294

**Published:** 2020-05-09

**Authors:** Éidín Ní Shé, Deirdre O’Donnell, Sarah Donnelly, Carmel Davies, Francesco Fattori, Thilo Kroll

**Affiliations:** 1School of Nursing, Midwifery and Health Systems, University College Dublin, Belfield, 4 Dublin, Ireland; deirdre.odonnell@ucd.ie (D.O.); carmel.davies@ucd.ie (C.D.); francescofattori3@gmail.com (F.F.); thilo.kroll@ucd.ie (T.K.); 2School of Social Policy, Social Work and Social Justice, University College Dublin, Belfield, 4 Dublin, Ireland; sarah.donnelly@ucd.ie

**Keywords:** assisted decision making, supported decision making, health system responsiveness, qualitative, person-centred care, public and patient Involvement

## Abstract

Objective: The Assisted Decision-Making (ADM) (Capacity) Act was enacted in 2015 in Ireland and will be commenced in 2021. This paper is focused on this pre-implementation stage within the acute setting and uses a health systems responsiveness framework. Methods: We conducted face-to-face interviews using a critical incident technique. We interviewed older people including those with a diagnosis of dementia (*n* = 8), family carers (*n* = 5) and health and social care professionals (HSCPs) working in the acute setting (*n* = 26). Results: The interviewees reflected upon a healthcare system that is currently under significant pressures. HSCPs are doing their best, but they are often halted from delivering on the will and preference of their patients. Many older people and family carers feel that they must be very assertive to have their preferences considered. All expressed concern about the strain on the healthcare system. There are significant environmental barriers that are hindering ADM practice. Conclusions: The commencement of ADM provides an opportunity to redefine the provision, practices, and priorities of healthcare in Ireland to enable improved patient-centred care. To facilitate implementation of ADM, it is therefore critical to identify and provide adequate resources and work towards solutions to ensure a seamless commencement of the legislation.

## 1. Introduction

There is now a formal recognition within health and social care systems that patients should be assisted in the decisions about their care [1,2,3]. Following this shift in 2015, the Assisted Decision-Making (ADM) (Capacity) Act was enacted in Ireland [4,5,6]. The Act gives effect to two international Conventions: the *UN Convention of the Rights of Persons with Disabilities* (2006) and the *Hague Convention on the International Protection of Adults* (2000) [3,5,6]. It is a significant piece of legislation providing the reform of the law regarding adults (18+) who require, or who may require, assistance in exercising their decision-making capacity [6]. The *UN Convention of the Rights of Persons with Disabilities* (2006) uses the term ‘supported decision making’ (SDM). The term has specific legal meaning placing the relevant person at the centre of decision making. SDM also outlines the provision of appropriate assistance to maximise the decision-making capacity of a relevant person. If deemed necessary, a person can name trusted people to assist them. Across legislative jurisdiction (such as in Australia, Canada, Israel, the United Kingdom and Germany) and in the literature [5,6], various terms have been used such as ‘supported decision making’, ‘active decision making’ and ‘assisted decision making’ (ADM). These terms are not interchangeable when referenced to the legal frameworks available to the person to make decisions within a particular jurisdiction. This paper uses the term ADM as outlined in the Irish act.

The purpose of the 2015 ADM Act, as it applies to health and social care, is to promote the autonomy of persons concerning their treatment choices, to enable them to be treated according to their will and preferences and to provide healthcare professionals with important information about persons and their choices in relation to treatment [4,5]. Under the Act, there is a statutory presumption that all individuals have decision-making capacity, with functional capacity tests used to determine capacity within a specific time and in relation to a particular issue and context [4]. The Act does not stipulate that incapacity is due to a medical cause and could also apply to those with intellectual, developmental and psycho-social disabilities. This also includes dementia and acquired brain injuries as well as those with fluctuating or temporary capacity [7,8]. The Irish ADM legislation does not contain any reference to the ‘best interests’ standard that has provided the basis for decision-making in other jurisdictions [5,7,8]. A statutory framework is outlined in the ADM act of tiered decision supports appropriate to the level of decision-making capacity and introduced statutory Advance Healthcare Directives into law [9]. The ADM legislation sets out a complicated structure for implementation within the health and social care system [5,6]. Relevant codes of practice for healthcare are in development by the Health Service Executive for publication in early 2021 [6,8].

### Gaps in the Evidence Base

Internationally, the evidence points to a gap of best practice in supporting ADM implementation within health and social care delivery [8,9,10,11,12,13]. Donnelly notes that the ADM legislation is ‘highly complex and at times impenetrable, even for experienced lawyers’ (p. 6) [6]. She argues it will require a significant focus on education to develop trust and understanding [6]. Ensuring that the healthcare system is responsive to the legislation is critical particularly as the implementation of ADM may require new approaches to care [2,3,14,15,16]. It is within this context that we designed the ‘Promoting Assisted Decision-Making within Acute Care Settings’ (PADMACs) study. The project is co-designing an educational tool which will promote understanding of ADM among HSCPs working in acute care settings and encourage their adoption of this understanding into their care planning with older people [4,8,9]. Our realist review of the literature focused on identifying mechanisms to support healthcare staff implementing ADM in practice [8]. The review recognised that the ADM Act is a complex clinical intervention involving multiple elements across various healthcare settings, organisations, professions and care sectors [8]. The broader implementation science literature stresses that healthcare staff can feel overburdened when expected to adopt legislative changes in environments that are not adequately resourced or supportive for the change in practice [17,18].

There has been no commencement of the health and social care provisions of the ADM Act. This is anticipated by 2021 [6]. The aim of this paper is focused on this pre-implementation stage of the ADM Act. We explore how responsive the Irish health system is currently to the implementation of the Act. Our interest in responsiveness is significant, given that the implementation science literature notes the challenges of preparing for implementation of completed policies and these challenges are often overlooked in the literature [9,15]. Responsiveness is now seen as a key characteristic of effective health systems [19,20]. This paper uses a health system responsiveness conceptual framework to presents the results of semi-structured interviews that were undertaken as part of the PADMACs study (Figure 1) [15].

At the core of a health systems responsiveness framework are the expectations from the public (individuals, families and communities) and other health system actors (policymakers, managers and service providers) of how individuals should be treated [15,19,20]. The focus of the framework is to ensure a health system is adaptive, and that it is responsive to needs [19]. Responsiveness is high in health systems when all staff have adequate autonomy, flexibility and resources to identify and adapt to the needs and expectations of individual users [19]. Globally, a health system’s responsiveness is now a consideration in assessing the quality of service provision [21]. The framework provides a lens to help us understand how responsive the Irish health system is to the ADM Act and guide our focus to any actions that might help to strengthen implementation.

## 2. Materials and Methods

### 2.1. Design

This was an exploratory qualitative study using face-to-face audio-recorded semi-structured interviews with follow-up validation of findings. The study participants were older people, including those with and without a diagnosis of dementia, family carers and health and social care professionals (HSCPs) working in the care of the older person in two acute hospitals. A narrative approach was adopted which generated rich story-based data in which the participants narrated their experiences and reflected upon what those experiences meant for them and for the care planning process [22]. This study is reported in line with the consolidated criteria for reporting qualitative studies [23].

### 2.2. Research Team Composition

The core research team consisted of six people. Five of the researchers have extensive experience in qualitative research, such as writing several articles using the methodology, regularly attending qualitative research-related lectures and seminars, and teaching lectures. One of the researchers has a clinical background in social work and another has a clinical background in nursing. One researcher is a professor of health systems, three are lecturing in health systems, nursing and social work and two are researchers. The project team members have experience of researching with vulnerable sections of the population including those with disabilities, communication impairment, those with a diagnosis of dementia and survivors of elder abuse. The team were guided by best practice, as underpinned by the Assisted Decision-Making Act (2015) in ensuring that accommodations and supports are in place which maximised the capacity of all potential participants to provide informed consent to participate in the interviews or follow-up validation groups.

### 2.3. Sampling and Recruitment

Purposive convenience sampling of health and social care professionals (HSCPs) delivering care to older patients was performed in two large level four urban academic teaching hospitals in Ireland. Project steering committee collaborators, who are health care professionals from each of the hospitals (three consultant geriatricians, one advanced nurse practitioner and the end-of-life care coordinator), assisted with participant recruitment. These committee members undertook broad dissemination of information about the project and acted as a link between the research interview team (FF, SD, DO’D and ÉNS) and potential participants. The core research team attended multi-disciplinary care-of-the-older-person team meetings and journal club sessions in the two sites and presented on the PADMACS study to enable dissemination of project information and facilitate recruitment. HSCPs who expressed interest were emailed a participant information sheet and invited to participate. A follow-up reminder email was sent after two weeks if no response was received.

Purposive convenience sampling was also used to recruit older people including those with or without a diagnosis of dementia and family carers. Recruitment was facilitated via our public and patient representative organisations Family Carers Ireland, the Alzheimer’s Society of Ireland and SAGE Advocacy. These organisations disseminated information, both in written form and orally, about the study within their existing membership. Each organisation promoted the study via different forums (member meetings, emails, newsletters, social media, text message and via word of mouth). Inclusion criteria were those who self-identified as older who had a recent experience of an acute care admission who wished to share their experience. For family carers the purposive recruitment focused on those with a recent experience where they had accompanied an older relative/friend to hospital and who wished to share their experience of assisting their relative or friend.

The gatekeeper organisations facilitated the communication of information regarding consent to the participants as well as reinforcing the right to withdraw from the study. The production of written communication material followed the National Adult Literacy Guidelines (NALA) for accessible communication. An information leaflet was provided to the potential participants by the organisation with contact details for the research team. Potential participants were instructed to contact a member of the research team (DOD) or the gatekeeper organisation in order to opt into study participation. Once participants indicate their willingness to take part in the research, they were provided with a consent form to read in their own time and share with a nominated decision-making assistant. They were offered support in understanding the consent form by both the organisation to which they belong and a member of the research team (DOD). They were offered the opportunity to have a discussion with a member of the research team as well as their designated decision-making assistant at any point prior to or while giving consent to participate.

### 2.4. Informed Consent and Data Management

The study was approved by the University College Dublin Human research ethics committee in 2018 (REC reference LS-18-73-ODonnell). All participants prior to the interview received written information before agreeing to participate. The interview team gave time at the start for all participants to read the information and ask any questions. At the beginning of the interview an explanation of the PADMACs study was provided verbally by the core interview team (FF, SD, DO’D and ÉNS). Following agreement from all participants and sign off via the consent sheet, an unobtrusive digital audio-recording device was used for all interviews. Verbal and written consent was taken at the start of the recording. All audio-recordings were transcribed verbatim, using station notation by an experienced transcription agency. The management of research data has been in line with requirements set out under the European Union’s general data protection regulation and Irish data protection legislation.

### 2.5. Data Generation

Interviews took place between January and June 2019. Interviews with HSCPs were undertaken in the hospital sites A and B in either personal offices or in uninterrupted private spaces. The interviews undertaken with older people, including those with a diagnosis of dementia and family carers, were undertaken across Ireland in locations as agreed with the participants (their homes, hotels, meetings rooms, and at the university). Additional time and breaks were allocated for these interviews.

Three separate interview guides for older people, family carers and HSCPs were developed by the research team using Flanagan’s critical incident technique (Appendix A). The technique is a systematic, inductive and flexible qualitative research method [24]. It is a methodology used to reconstruct experiences by collecting and analysing data with the aim of providing solutions to practical problems [25]. The anonymous interviews explored participants’ understanding of ADM during a recent acute care experience and captured suggestions from them of what is required to embed ADM in practice. For older people with a diagnosis of dementia we consulted with the Alzheimer’s Society of Ireland who advised that some interviewees may have memory challenges. With their guidance we created a scenario as a prompt if required by interviewees. Only one interviewee required this innovation.

### 2.6. Data Collection and Analysis

In the first phase of analysis, inductive thematic coding of the data was undertaken via a licensed version of NVivo 12 Pro which enabled the large volume of data to be organised thematically. The data was organised into conceptual themes through a three step process: (1) at the lowest order, basic themes were initially extracted through NVIVO coding, (2) the NVIVO codes were aggregated into more abstract organisation of themes and, finally, (3) macro themes were constructed which brought together lower-order themes. All the interviews were coded into NVIVO lower-order codes by one researcher (FF), and these were then verified by a second researcher (DOD). Discrepancies between the two researchers were discussed and resolved at fortnightly meetings with the broader research team (CD, ÉNS, SD, TK).

Macro themes (level three) were agreed by the two researchers (FF and DOD) and validated by the research participants. Presentations of initial thematic findings and a tentative macro thematic structure were made to each of the participant cohorts. Journal club sessions were run in each of the hospital sites and participants were invited to attend and contribute to the validation of the data collected from HSCPs. A summary document of the thematic analysis of the family carer and older people’s data was sent to each of the non-HSCP interviewees and follow-up phone calls or home visits were made to discuss the tentative findings and gather their feedback. Upon completion of the respondent validation process, a broader research team meeting was held in which participant feedback was discussed and the macro-thematic structure of the findings was consolidated.

For the focus of this paper, a framework analysis was used [26] in a further round of coding undertaken by a second researcher (ÉNS). This involved transposing the themes identified from the original data coding onto the health system responsiveness conceptual framework. This framework describes three main domains: context, experiences and responses (Figure 1). Data counts have been illustrated with words such as ‘a majority’ or ‘several’ in keeping with the narrative style of the results. Pseudonyms are used to present the findings to protect the confidentiality of participants.

## 3. Findings

### 3.1. Background of Interviewees

The convenience sampling for participants resulted in 39 interviews conducted with older people including those with a diagnosis of dementia (*n* = 8), family carers (*n* = 5) and HSCPs working in the acute setting (*n* = 26). Table 1 provides a breakdown of participants’ characteristics. Different numbers of participants are represented in each category due to the consecutive nature of the convenience sampling which ran for 6 months.

### 3.2. Health and Social Care Professionals’ Characteristics

A total of 26 health and social care professionals working in the care of the older person responded to the study invitation (doctors, nurses and allied health professionals (dieticians, speech and language therapists, social workers, occupational therapists and physiotherapists). All were working in an acute setting for five years and more. Interviews ranged from 15 min to 1 h 10 min.

### 3.3. Older People Including Those with a Diagnosis of Dementia and Family Carers Characteristics

A total of 13 participants responded to the invitation. The older people with a diagnosis of dementia (*n* = 4) had a recent diagnosis (5 years or less). The older people (*n* = 4) were all over 65 years of age and had a recent admission in an acute hospital. Family carers interviewed were caring for partners (*n* = 1); siblings (*n* = 2) and parents (*n* = 2). The reason for care ranged from Parkinson’s disease (*n* = 1) to dementia (*n* = 1), cancer (*n* = 1) and intellectual disability (*n* = 2). Interviews ranged from 30 min to 1 h 50 min and were carried out in locations across the country as agreed with by the participants.

### 3.4. Analysis Results

The themes generated from the interview data were coded using the health system responsiveness conceptual framework (Figure 1) [15] involved the following three main domains:Context (historical, political, social economic)Experiences of how people, including health care staff, interact with their health systemHeath systems response to the interaction

A total of 10 codes were derived and sub-categorised using the health systems responsiveness conceptual framework (Table 2).

These are described in more details with the use of de-identified interviewee quotes below.

### 3.5. Context (Historical, Political, Social, Economic)

The majority characterised the context of the health care system under significant resource and time constraints. Older people and family carers noted that they frequently encountered a healthcare system that is constantly very busy and overwhelming for them. Síofra explains her recent experience of an emergency department comparing it to a war zone. She felt she was not a priority:

“I said I don’t want to lie in a corridor on a trolley, but the emergency department was very, very, very busy. I have never been in a war situation, but it felt like a war zone that is the best way to put it. There was people running there, people crying, there was a breeze, and once you were lying there and you didn’t seem distressed or you didn’t seem like you needed urgent care then you were ok. You are not a priority, and it’s absolutely not the doctors and nurses’ fault. I have no idea how any of them work in accident and emergency. It is absolutely scandalous what they have to do.” (Síofra, Older person with Dementia)

Interviewees also outlined their experience of the acute care setting as a difficult and an unacceptable environment. Blame was not attributed to frontline staff. One older person with a diagnosis of dementia outlined her experience of being frequently moved during a recent hospital admission. She felt her diagnosis meant that she was not listened to:

“Hospitals are frightening places. They are disabling place I was not just being listened to because of a diagnosis with early onset Alzheimer’s but to understand that I could make decisions for myself. I was moved five times in eleven days. This was certainly not the fault of the sister on the ward who was most helpful to me. But the beds manager who obviously had no training in dementia just decided the move had to happen. I cannot tell you how frightening that is” (Liana, Older person with Dementia).

The constant busyness of the acute environment had normalised a lack of time for patient engagement. Frontline HSCPs expressed their own frustrations over not having enough time to engage with patients.

“When you are working in a hospital you are always constrained by the time that you have with a patient in relation to anything that you do” (Sean, HSCPs).

Healthcare staff perceived the acute care context as being ill-prepared to implement the ADM in practice. Many questioned how the ADM Act could be put into practice considering the pressures and resource constraints present in the current system. One nurse shares her thoughts:

“In the context of an acute hospital, it is a medical model that is overcrowded, that they just want quick discharges, and it’s very hard to work on that model.” (Jane, Nurse).

Added to the lack of time and the huge pressure currently in the system, all healthcare workers who were interviewed expressed frustration at the lack of resources available to them. One doctor noted how most patients wanted to return home, but very little options or adequate resources were available to enable that:

“People shouldn’t be going to nursing homes against their will and it’s deprivation of liberty and all that. Actually, we don’t have alternatives to offer them. The majority of people want to go home, and you know, but you actually can’t provide them with a service. I don’t know if there will ever be anywhere that service there” (Áine, Doctor).

Several noted that when they had taken the time and ascertained the will and preference of a patient, very few options were available to them. This created vulnerabilities for them when they engaged with patients.

“What bothers me most is the disparity between the choices that people have or actually don’t have. The choices that they feel they have but actually they don’t. And how we feel like the bad guys because there is only the one option of the nursing home. So, it’s certainly not ideal.” (Tomás, Doctor).

Allied healthcare staff in one hospital outlined how a command and control hierarchy was in place for decisions. Frequently, it was the consultant who determined a patient’s decision-making capacity without the input of other staff members.

“It’s still kind of not considered culturally here or appropriate for me to say that the man has capacity, it’s still very much viewed as that’s the consultant’s decision” (Daithí, HSCPs).

Staff felt they could not speak up and challenge senior doctors. The culture in the hospital had embedded this practice:

“We have an old-fashioned culture at this hospital. And that is just the way it is, it’s very medical heavy dominant in decision making” (Gráinne, HSCPs).

### 3.6. People’s Interaction with their Health System

Communication was identified as a significant sub-theme within people’s interaction with the healthcare system. One family carer whose sister has an intellectual disability expressed her frustration that doctors were dismissive of her sister.

“He was so dismissive to us when we asked for further information…So I ended up ringing my two brothers to come up and we demanded a meeting with the doctor” (Meadhbh, Family Carer).

Interviewees explained that they had to really push to have their voices heard and to have their wishes acted upon by healthcare staff.

“I would say an awful lot of people are afraid of the consultants, afraid of the doctors, terrified to ask them anything in case oh if I say something you know they you know they will have it in for me you know” (Larry, Older Person).

Due to a lack of integrated data systems, several noted that they had to constantly repeat case history. Older people and family carers explained that when stressed, they could forget specifics, or became worried about leaving important information out. Often this made them very anxious. An older person with a diagnosis of dementia expressed frustration that their diagnosis was not on file despite having a recent admission in the same hospital. There were also challenges for hospital staff when patients had several previous admissions to other local hospitals and there was no access to information sharing across the health system. Examples were also given of healthcare staff doing all they could within the constraints of a very busy work environment to enable communication and information sharing.

“We had a picture book with her favourite foods with her family members with pictures so that she could point if she wanted to speak to anyone” (Amber, HSCPs).

The lack of privacy for conversations within the acute setting environment was also noted. Many conversations often occurred in busy wards or in corridors and often did not include the patient.

“Sometimes you are talking to families in a corridor it’s far from ideal and the first thing you have to say is firstly I am really sorry we are having this chat in a corridor there just is nowhere else”(Peter, Nurse).

A doctor explained the additional time and resources it would take to include a patient:

“So, if we wanted to take him down to the family room, which is always interrupted every five minutes when you are sitting there, we would have had to have a carer put him in his supported chair you’d be talking about a thirty-minute ordeal to have a quick chat with them. So, a lot of the times no, it was in the ward at the bedside or else I was catching up with his wife you know sort of as we do on the corridors and things.” (Jayne, Doctor).

Information sharing with community partners (such as GP’s; public health nurses) was often difficult for HSCPs as they tried to link up services. Without an integrated electronic heath record, they were often cold calling colleagues in the community to request supports for a patient.

### 3.7. The Health Systems Response to the Interaction

Decisions made by patients that were deemed to be ‘unwise’ by some healthcare staff will require a shift in practice for some.

“Even if it may seem unwise or something a bit unsafe but if someone has that decision and they have that capacity to make that decision then I think that there needs to be kind of a shift in mentality to empowering them” (Fernando, HSCPs).

Whose role it was to assist older patients in their decision making was also reflected upon. There was a recognition that it should be everyone’s job, but it was often deemed to be carried out by geriatricians.

“A lot of our geriatric patients aren’t under geriatricians…Everyone is under time pressure and there is a busy caseload and you are constantly prioritising and deprioritising. And I mean everyone is trying their best but often you can’t always get a patient seen by a geriatrician for a formal geriatric assessment” (Regina, HSCPs).

Allied healthcare staff expressed concern that the ADM legislative rollout was very doctor led within their hospitals. They felt that the governance should be inclusive of all staff to ensure adequate implementation.

The lack of basis or adequate resources to support patients and healthcare staff was a frequent occurrence.

“They hadn’t got the new air mattress. So, I had to stay all that night and get a nurse every hour in to help me to turn him because … And I’d say by ten o’clock, I was dying on my feet” (Áine, Family Carer).

“We’d difficulty getting an interpreter in to discuss with him about his care needs” (John, HSCPs).

The disparity of options between those who had the means to pay for their own home care versus those who were dependent on publicly funded system supports was very clearly stressed in interviews with healthcare staff.

“We weren’t able to get him home and mostly because the type of care package he would need doesn’t exist. So that was difficult.” (Patricia, HSCPs).

“She was able to afford to pay for essentially twenty-four-hour care. Which isn’t a position that a lot of people find themselves in” (Tina, Doctor).

For healthcare staff, this gap was all too frequent:

“We can be left in a very difficult situation whereby we all want the same thing, but the services are not available and therefore somebody cannot go home. So, there’s a huge gap between what people want and what they can have, unfortunately” (Tómas, HSCPs).

Without flexible and patient-centred resources, most interviewees felt that it would be difficult to implement the ADM Act.

## 4. Discussion

Health system responsiveness indicates the ability of a health system to meet the population’s legitimate expectations regarding non-medical and non-financial aspects of their care process and experiences within it [27]. Levels of responsiveness are high when staff have the opportunity, mechanisms, and funding to respond to patient’s needs. The use of the health system responsiveness framework provides a useful oversight to capture the performance of the Irish health system as the new ADM legislation is about to be commenced [15,27,28]. The framework as outlined in this paper can provide feedback and understanding to service users, implementers and policy makers as to what needs to be prioritised for implementation of the ADM act.

Ireland, as with many health systems, has an ageing population with the increasing prevalence of dementia and complex support needs [4,6,7,8]. The interviews presented in this paper raise a concern about how health and social care services will respond to the increasing numbers that will need assistance with their decision making once the ADM legislation is commenced [5,6,8,9]. When aligned with the experiences within a health systems responsiveness framework, the interviews indicate the overall quality of dignity, autonomy, confidentiality, attention, access to networks, quality of amenities and choice of provider communication trust is currently low. The interviews point to a healthcare system that is currently under significant resource and time strain. This leads us to question the responsiveness of the health system to implement the ADM act. Frontline staff are doing their best within these pressures, but they are often impeded from realising the will and preference of their patients by scarce resources and the rationing of health and social care services. The environment of the acute setting lacks private spaces for conversations. For older people and family carers, many feel that they must be very assertive to have their preferences considered, with some often afraid of what the perceived consequences would be. All expressed concern about the resource strain on the healthcare system. Conversations are often occurring in environments which are not private or patient centred. The interviews also highlighted clear inequities between those who can afford to pay for additional home care supports versus those who are dependent on an under-resourced and inflexible public system.

Legislation frameworks that support ADM are present across several countries around the world [1,2,9]. The literature has stressed that in order to enable assisted decision-making, health care systems need support in reorganising how their services are delivered and ensuring that staff are supported via ongoing learning [2,8]. Organisations that fail to provide leadership and resources put additional pressures on front line healthcare staff and increase the likelihood of implementation failure [10,28]. Several of the interviewees who are health and social care professionals pointed to hierarchical cultures embedded within their organisations. Many felt that their skills and insights on capacity may be overlooked in these contexts. The literature is clear in stressing when little evidence of a strong culture of interprofessional collaboration is enabled or where the biomedical model is dominant, then ADM is not implemented [10].

There is agreement that the ADM legislation is complex with substantial challenges highlighted in the literature and from the interviews presented in this paper on its implementation [5,6,8,21]. The ADM Act does, however, resonate with quality care and person-centred healthcare [9,10]. As we move to more personalised approaches in health and social care, the research highlights growing inequalities [6,28,29]. Differential access to health and social care services can occur because of geographical, economic and cultural reasons [30]. Within Ireland, the evidence points to significant regional inequalities in the distribution of primary, community and long-term care services and a healthcare system under severe strain [31]. This raises concerns about what additional supports will be available across the country to realise the ADM Act [6,9]. There is agreement within the literature that big legislative and policy changes are dependent on what additional supports and resources are provided to enable implementation, and it is where often a wide gap exists [9,11,12]. A recent example in Ireland of the implementation of the legislation of abortion in some circumstances in January 2019 shows that additional supports and resources varied across the country [32]. Senior medics have outlined that there has been a lack of training, resources and other supports following commencement of the Health (Regulation of Termination of Pregnancy) Act in 2018. Certain parts of the country have not introduced a service leaving healthcare staff worried they would be blamed for inevitable failing and inequities in access and variations in care [32]. We have outlined in this study clear implications for practice. The ADM legislation will be commenced in late 2020 with codes of practices forthcoming. Without aligned, flexible resourcing and supports such as data sharing, ongoing education for all healthcare staff and environmental restructuring to enable privacy within the acute hospital, there will be challenges in fully implementing health system responsiveness to ADM.

### Strengths and Limitations of the Work

This study has several limitations. This study was limited by being focused on older persons services, and therefore, the findings may not be transferable. We did not speak to communities or policymakers, as included in the health systems responsiveness framework (Figure 1). This decision was taken as the ADM legislation has not yet been commenced. Another limitation is that the interviews with healthcare staff were undertaken in two large acute urban settings. Further research should focus on staff working in rural and community settings. There is growing recognition of the need to understand the implementation of legislation [6,8,9,11,12,30]. This work contributes to that literature. This research is unique in gathering diverse experiences of ADM and has several key strengths. It is the first known qualitative study on the Irish health system’s responsiveness to ADM. The use of semi-structured in-depth interviewing allowed for detailed exploration of emergent issues.

## 5. Conclusions

The commencement of ADM provides an opportunity to redefine the provision, practices, and priorities of healthcare in Ireland to enable improved patient-centred care. Responsiveness is high in health care systems when health and social care staff members have enough skills, autonomy, flexibility and resources to identify and adapt to the needs and expectations of older people and family carers. Many challenges raised in this study are relevant to the health system responsiveness framework and provides a useful oversight to enable future planning and supports. To facilitate implementation, it is therefore critical to identify and provide adequate resources that align with an implementation plan and work towards solutions to ensure a seamless commencement of the legislation.

## Figures and Tables

**Figure 1 ijerph-17-03294-f001:**
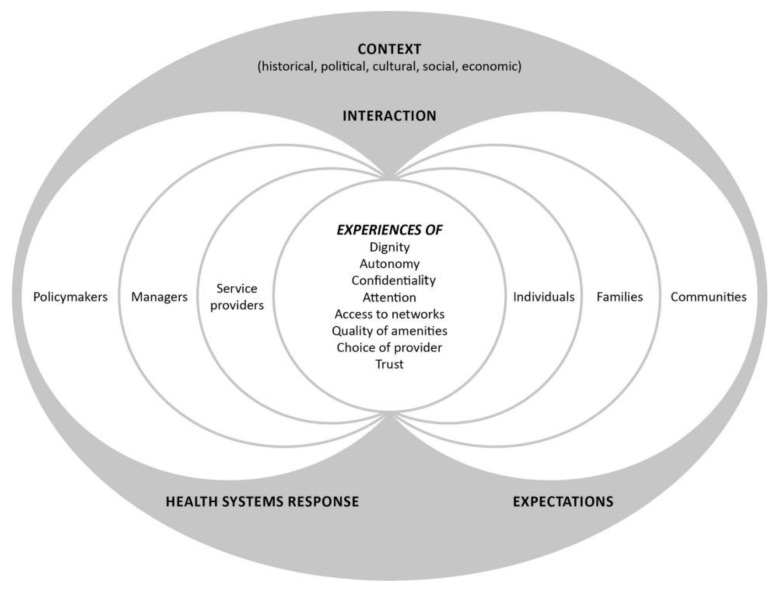
A conceptual framework for health systems responsiveness (Reproduced from Mirzoev and Kane, 2017) [15] Creative Commons Attribution (CC BY 4.0) license.

**Table 1 ijerph-17-03294-t001:** Background information of research participants (*n* = 39).

Group	Breakdown	Male	Female
Hospital Staff 26 participants	Doctors *n* = 10	1	9
	Nurses *n* = 3	1	2
	Allied Health *n* = 13	1	12
Older People (over 65) 8 participants	Older People *n* = 4	4	0
	Older People with a diagnosis of Dementia *n* = 4	2	2
Family Carers 5 participants	Family Carers *n* = 5	0	5
Total		9	30

**Table 2 ijerph-17-03294-t002:** Structure of the analysis results using health system responsiveness conceptual framework and the main content.

Category	Sub-Category
1. Context	1.1 Resource and Time constraints
	1.2 Acute care setting as a difficult environment
	1.3 Command and control hierarchy
2. Experiences	2.1 Communication
	2.2 Repeating details
	2.3 Lack of Privacy
	2.4 Challenges of information sharing across the health care system
3. Response	3.1 Accepting an ‘unwise’ decision
	3.2 Doctor led Implementation
	3.3 Disparity of options

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
