# Peer review of "“What Bothers Me Most Is the Disparity between the Choices that People Have or Don’t Have”: A Qualitative Study on the Health Systems Responsiveness to Implementing the Assisted Decision-Making (Capacity) Act in Ireland"

_ijerph, 2020, doi:10.3390/ijerph17093294_

Round 1
Reviewer 1 Report
The aim of the Promoting Assisted Decision-Making within Acute Care Settings’ (PADMACs) study is to assess how responsive the Irish health system is to the ADM Act.
Assisted Decision-Making (ADM) promote the autonomy of persons concerning their treatment choices, to enable them to be treated according to their will and preferences and to provide healthcare professionals with important information about persons and their choices in relation to treatment.
The study consists in face-to-face interviews using a critical incident technique in 8 older people (including those with a diagnosis of dementia), 5 family careers and 26 health care professionals working in the acute setting.
This is a very interesting and much-needed study, but the selection and number of participants is clearly insufficient to be representative.
The selection of respondents carries a significant bias. “Participants who expressed interest were emailed a participant information sheet and invited to participate”.
It is striking that the authors took 6 months to carry out 39 surveys (8 older people, 5 family careers and 26 health care professionals). Sex and type of work greatly influence the responses. Table 1 show that there are groups of 1 participant. This invalidates any analysis of results.
In table 1 there is an error. There are only 30 women
In line 391 there is a number that I suppose is a bibliographic citation but poorly formatted.
The authors forget that the main limitation of the study is the small number of respondents. It is essential to perform a sample size calculation and delimit the study population so that the study could be representative for a specific population (patients or caregivers or doctors or auxiliaries ...). With these scarce data, the authors have been unable to present a single table of results. Therefore, the conclusions cannot be based on any results.
It is a very well prepared study, but without results. It can be published as a study proposal and results can be presented in a second publication.
Author Response
On behalf of the authors I wish to thank you for the feedback on our paper. I attached the changes and responses we undertook. Regards, Éidín.

Reviewer 2 Report
Overall a well written paper.
The introduction / background gives a good perspective on policy, international aspects and then focusing on Ireland.
Line 67: Decison supports as a phrase was duplicated.
The realist approach seemed appropriate.
Figure 1 - in relation to a responsive health care system was very good.
There was appropriate ethical review of the study.
Could the authors please describe how the participants with dementia or intellectual impairment were able to consent to the study?
Line 228 Listened rather than listed
The authors picked up on the challenge of wanting to send patients home (but resources not sufficient) compared to sending patients to a nursing home (which often the patients and families did not want)
The framework of context, experiences and health service response was a good one to outline the results.
Line 360 Every in stead of ever ..
The limitations of the study were outlined.
Perhaps the authors could more explicitly mention that the study included patients with dementia and intellectual impairment, in the conclusion.
Author Response
On behalf of the authors, I wish to thank you for the feedback on our paper. I attached the changes and responses we undertook. Regards, Éidín.

Reviewer 3 Report
The article presents an important research topic. I generally assess the study’s design and presentation positively, with the following remarks:-
authors should additionally present the available studies presenting this problem from a theoretical perspective, with identification of specific factors identifying the analyzed issues-
I suggest describing how the research sample is constructed (selection of participants)-
How is the data collected and organized (encoding)?Describe the method of data encoding used to identify factors of the selected narration content reflecting experience related to problems presented in the research of Assisted Decision-Making (ADM).
I propose, in this narrative study, creating a matrix in the form of a table, which will present the context of experience, the subject of experience, and the spectrum of experiences.
The conclusions must be organized by analysing selected parts of the narration content organized in the table.
Author Response
Thank you for reviewing this work and providing feedback on our paper. I attach our changes and responses. Regards, Éidín

Round 2
Reviewer 1 Report
I believe it can I believe it can be published with the changes that have been made.pI believe it can be published with the changes that have been made.ublished with the changes that have been made.